# Innovative Therapeutic Strategies for Myocardial Infarction Across Various Stages: Non-Coding RNA and Stem Cells

**DOI:** 10.3390/ijms26010231

**Published:** 2024-12-30

**Authors:** Bingqi Zhuang, Chongning Zhong, Yuting Ma, Ao Wang, Hailian Quan, Lan Hong

**Affiliations:** 1Department of Physiology and Pathophysiology, College of Medicine, Yanbian University, Yanji 133002, China; 15957661009@163.com (B.Z.); 13944948637@163.com (C.Z.); 15981188912@163.com (Y.M.); 2Experimental Teaching Center, College of Pharmacy, Yanbian University, Yanji 133002, China; 0000008000@ybu.edu.cn

**Keywords:** stem cell, non-coding RNA, microRNA, long non-coding RNA, heart disease, cell therapy, exosome therapy

## Abstract

Myocardial infarction (MI) is a highly challenging and fatal disease, with diverse challenges arising at different stages of its progression. As such, non-coding RNAs (ncRNAs), which can broadly regulate cell fate, and stem cells with multi-differentiation potential are emerging as novel therapeutic approaches for treating MI across its various stages. NcRNAs, including microRNAs (miRNAs) and long non-coding RNAs (LncRNAs), can directly participate in regulating intracellular signaling pathways, influence cardiac angiogenesis, and promote the repair of infarcted myocardium. Currently, stem cells commonly used in medicine, such as mesenchymal stem cells (MSCs) and induced pluripotent stem cells (iPSCs), can differentiate into various human cell types without ethical concerns. When combined with ncRNAs, these stem cells can more effectively induce directed differentiation, promote angiogenesis in the infarcted heart, and replenish normal cardiac cells. Additionally, stem cell-derived exosomes, which contain various ncRNAs, can improve myocardial damage in the infarcted region through paracrine mechanisms. However, our understanding of the specific roles and mechanisms of ncRNAs, stem cells, and exosomes secreted by stem cells during different stages of MI remains limited. Therefore, this review systematically categorizes the different stages of MI, aiming to summarize the direct regulatory effects of ncRNAs on an infarcted myocardium at different points of disease progression. Moreover, it explores the specific roles and mechanisms of stem cell therapy and exosome therapy in this complex pathological evolution process. The objective of this review was to provide novel insights into therapeutic strategies for different stages of MI and open new research directions for the application of stem cells and ncRNAs in the field of MI repair.

## 1. Introduction

Myocardial infarction (MI) remains one of the leading causes of mortality worldwide, with the limited regenerative capacity of cardiomyocytes (CMs) posing a significant barrier to cardiovascular disease recovery. MI typically arises from an imbalance in cardiac oxygen supply and demand, often due to a vascular occlusion, which marks the initial stage of the condition. During this early stage, the primary clinical challenge is the restoration of adequate blood and oxygen supply to the heart to prevent further deterioration [1]. Without a timely intervention, the infarcted area expands as more CMs undergo apoptosis, transitioning the disease to a progressive phase characterized by widespread myocardial cell death and increasing the infarct size [2].

Although traditional treatments, such as thrombolytic therapy, can alleviate vessel blockage and improve perfusion, they may also trigger secondary cardiac injuries, including reperfusion damage and ventricular remodeling. At this stage, therapeutic strategies need to focus on supplementing functional CMs to replace the lost myocardium and mitigate the adverse effects of reperfusion-induced injury.

In the terminal stage of MI, the situation becomes even more critical. Extensive myocardial damage leads to irreversible heart failure, where the only viable option may be heart transplantation [3]. However, the success of transplantation is often hindered by immune rejection, making immunosuppressive management a pivotal focus.

Each stage of MI presents distinct challenges: from restoring oxygen supply and preventing acute damage in the early stage, to addressing cardiomyocyte loss and secondary injuries during disease progression, and ultimately managing heart failure and immune rejection in the terminal phase. These complexities underscore the urgent need for novel therapeutic approaches tailored to the specific pathophysiological characteristics of each stage of MI.

Stem cells, with their multidirectional differentiation potential and paracrine functions, are regarded as “seed cells” superior to traditional pharmacological therapies, and constitute one of the important means for treating heart injury. By transplanting induced pluripotent stem cells (iPSCs) or mesenchymal stem cells (MSCs), it is possible to replenish the normal CMs lost in the injured area, regulate immune responses, and promote the recovery of cardiac function. Meanwhile, stem cell-derived exosomes, which contain a variety of functional molecules and exhibit low immunogenicity, have garnered extensive attention in recent years and emerged as a highly promising therapeutic option [4,5]. Non-coding RNAs (ncRNAs), as intracellular gene regulatory factors, not only directly participate in the repair of an infarcted myocardium and guide stem cell differentiation, but also serve as key functional components in exosomes to facilitate recovery from heart injury. Within the genome, protein-coding genes account for only a small proportion, while the majority of genes reside in non-coding regions. The RNAs transcribed from these regions, such as microRNAs (miRNAs) and long non-coding RNAs (LncRNAs), regulate cell development and function [6]. MiRNAs and LncRNAs are primarily distinguished by their length: miRNAs are approximately 22 nucleotides in length, whereas LncRNAs exceed 200 nucleotides. MiRNAs can assemble with cytosolic proteins into RISC complexes, mediating mRNA degradation or inhibiting its translation, thereby achieving target gene silencing. In contrast, LncRNAs participate in complex gene expression regulatory networks by serving as signaling molecules, decoys, guides, or scaffolds [7].

Utilizing ncRNAs for the direct treatment of heart injury or in combination with stem cells and their exosomes to achieve myocardial repair offers novel research avenues for the treatment of MI. However, there is currently a lack of in-depth understanding regarding the specific roles and mechanisms of ncRNAs, stem cells, and their exosomes at different stages of MI. This review systematically categorizes the different stages of MI and comprehensively summarizes the ncRNAs that have regulatory roles in myocardial repair at each stage. It also explores the specific functions and underlying mechanisms of stem cell therapy and exosome treatment during the disease course, aiming to provide new insights into treatment strategies for each stage of MI and to explore new research directions for the application of stem cells and ncRNAs in myocardial repair.

## 2. The Role of ncRNA in the Treatment of the Initial Stage of MI

In the early stage of infarction, reduced blood flow to a portion of the myocardium leads to localized ischemia and hypoxia in the heart. At this point, the number of dead CMs is limited. If the diagnosis of patients with minimal ST segment changes and no increase in myocardial enzymes in the early stage of infarction is completed promptly, and the cardiac blood and oxygen conditions are improved as soon as possible, the deterioration of infarction can be avoided, significantly improving patient survival rates [8]. Unlike anticoagulant therapy or surgical revascularization, ncRNA-induced cardiac angiogenesis can fundamentally restore cardiac blood and oxygen conditions. For example, the direct injection of ncRNA into the infarcted heart can promote the proliferation of ECs in blood vessels, induce transplanted stem cells to differentiate into vascular cells, and protect vessels in the infarcted area from damage through exosomes, thereby facilitating angiogenesis (Figure 1).

### 2.1. The Role of miRNAs in Angiogenesis Following MI

Research has demonstrated that the introduction of specific miRNAs can significantly promote angiogenesis in infarcted myocardial regions. For example, introducing miR-21-5p into infarct zones in pigs upregulates the levels of VEGFA and PDGF-BB in endothelial cells (ECs), thereby activating phosphorylation pathways such as ERk1/2, FAK, P38, and AKT. This accelerates vascular formation and effectively reduces infarct size [9]. Similarly, miR-218 enhances EC proliferation, migration, and angiogenesis by suppressing HMGB1 expression, positively impacting cardiac functional recovery [10]. miR-216a, by inhibiting autophagy-related genes PTEN and BECN1, promotes the proliferation and migration of human microvascular endothelial cells (HMVECs), which is crucial for microvascular formation in infarct regions and delaying heart failure progression [11]. Additionally, overexpressing miR-210 significantly downregulates Efna3 mRNA and Ptp1b protein levels in HL-1 cells, thereby reducing CM apoptosis and creating favorable conditions for angiogenesis in the infarcted heart [12].

Not all miRNAs positively influence angiogenesis. Certain miRNAs may induce apoptosis in ECs, inhibiting proliferation and migration, thus impeding vascular regeneration. For instance, miR-92a obstructs EC proliferation and migration after injury by inhibiting ERK1/2, JNK/SAPK, and eNOS phosphorylation, and downregulating KLF4 and MKK4 expression. Suppressing miR-92a expression can instead promote endothelial regeneration and prevent post-thrombectomy re-narrowing, helping maintain normal oxygenation in the heart [13,14].

The exposure of endothelial progenitor cells (EPCs) to lipoprotein(a) [LP(a)] increases miR-221-3p levels, leading to SIRT1 downregulation and blocking the RAF/MEK/ERK signaling pathway, thereby reducing EPC proliferation and migration. However, inhibiting miR-221-3p expression enhances EPC resistance to LP(a)-induced damage, promoting angiogenesis and effectively lowering the risk of MI [15]. Following MI, the accumulation of miRNA-24 in ECs downregulates eNOS, exacerbating the infarct condition. The suppression of miRNA-24 expression can enhance EC survival and proliferation, facilitating capillary network formation in the infarcted region and alleviating cardiac pathology [16].

In a mouse model of acute MI, increased miR-26a expression in ECs inhibits the bone morphogenetic protein/SMAD1 signaling pathway, leading to reduced Id1 protein expression and increased levels of p21 (WAF/CIP) and p27. Anti-miR-26a treatment to inhibit miR-26a expression stimulates neovascularization in the infarct region, reducing the infarct area [17]. Hypoxic conditions post-infarction induce the transcription of miR-223-3p in cardiac microvascular endothelial cells (CMECs), which suppresses the RPS6KB1/HIF-1α pathway, downregulating VEGF levels and inhibiting MAPK, PI3K, and AKT expression. These changes hinder CMEC proliferation and migration, ultimately interfering with angiogenesis in the infarct region [18].

### 2.2. The Impact of LncRNAs on Angiogenesis in MI

Cardiovascular pathologies, especially arterial thrombosis and atherosclerosis, are significant triggers for MI. Throughout this complex process, LncRNAs demonstrate unique regulatory roles; some can alleviate arterial damage and promote angiogenesis, while others may inhibit vascular regeneration, thereby either slowing or exacerbating the progression of MI.

MALAT1, first identified in lung cancer tissue, exhibits extensive regulatory functions in cardiovascular regeneration. MALAT1 enhances EC division and promotes angiogenesis by upregulating S-phase cyclins CCNA2, CCNB1, and CCNB2, while downregulating cell cycle inhibitors p21 and p27Kip1 [19]. Additionally, MALAT1 effectively suppresses LPS-induced macrophage activation, reducing inflammatory cytokine release, thus aiding in atherosclerosis regulation and arterial repair [20]. Notably, MALAT1 interacts with another LncRNA, NEAT1, to collectively modulate the levels of inflammatory factors in arteries, further alleviating atherosclerosis [21]. As a competitive endogenous RNA for miR-26b-5p, MALAT1 promotes Mfn1 protein expression, inhibiting mitochondria-dependent apoptosis in CMECs, thereby enhancing angiogenesis and supporting microcirculatory repair post-infarction [22].

Beyond MALAT1, other LncRNAs also play essential roles in angiogenesis following MI. For example, H19 overexpression in arterial ECs inhibits STAT3 phosphorylation, which reduces the levels of intercellular adhesion molecule 1 (ICAM-1) and vascular cell adhesion molecule 1 (VCAM-1), promoting cell proliferation and delaying senescence to alleviate atherosclerosis [23]. Enriched in human vascular cells, SENCR binds to cytoskeleton-associated protein 4 (CKAP4) in ECs, preserving cell morphology and preventing vascular pathologies [24]. Furthermore, LncRNA ANRIL increases the phosphorylation levels of AKT and eNOS in ECs, promoting angiogenesis under ischemic conditions and alleviating the hypoxia–glucose deprivation (OGD) environment in the heart [25].

However, the impact of LncRNAs on cardiac angiogenesis is not universally beneficial. Some LncRNAs inhibit vascular regeneration. For instance, TUG1 deactivates HIF-1α, downregulating VEGF-α levels and hindering the regenerative capacity of ECs in post-infarction cardiac microvasculature [26]. LncRNA RP11-714G18.1 binds directly to its neighboring gene, LRP2BP, activating its expression, which subsequently downregulates matrix metalloproteinase (MMP)1 and suppresses EC migration [27], thus impairing angiogenesis. The upregulation of LncRNA PCAT19 increases N-acetylglucosaminyltransferase 2 (GCNT2) expression, ultimately inhibiting proliferation and angiogenesis in human coronary artery endothelial cells (HCAECs) [28]. Additionally, TTTY15 reduces cellular activity and migration range by inhibiting miR-455-5p and increasing JDP2 levels in CMs, negatively affecting cardiac angiogenesis [29].

### 2.3. Regulation of ncRNAs in Stem Cell Therapy for MI to Promote Angiogenesis in Infarcted Regions

In the early stages of MI, CM destruction results from ischemia and hypoxia caused by arterial occlusion. Studies have shown that MSC transplantation, modulated by ncRNAs, can stimulate angiogenesis at the site of injury, thereby improving oxygenation in the heart and slowing disease progression. Cellular studies demonstrate that overexpressing EC-specific miR-126 in transplanted MSCs enhances their secretion of angiogenic factors and resilience to hypoxia. In vivo experiments where MSCs overexpressing miR-126 were injected into infarcted myocardial regions showed increased levels of angiogenic factors and DLL-4 expression seven days post-injection. This resulted in improved myocardial blood flow and microvascular density nearing normal levels [30].

Additional studies have found that upregulating miR-21 in bone marrow-derived MSCs (BMSCs) elevates Bcl-2, VEGF, and Cx43 expression. Transplanting MSCs overexpressing miR-21 induces functional angiogenesis in ischemic myocardium while protecting CMs from apoptosis in the early stages of acute MI [31]. Xu et al. demonstrated that platelet-derived growth factor (PDGF)-BB in cell studies can upregulate miR-16-2 while downregulating miR-23b, miR-27b, and miR-320b in MSCs, subsequently activating transcription factor c-Jun via the phosphorylation of activator protein-1 (AP-1), promoting VEGF expression, inducing angiogenesis in the infarcted region, and restoring cardiac function following ischemia–reperfusion injury [32]. Furthermore, MSCs transfected with miR-126 in cell studies and then transplanted into the heart showed an upregulated expression of key genes in the angiogenic pathway, including ERK1, pERK1, AKT, and pAKT, significantly enhancing vascularization and cardiac function in the infarcted regions of mouse hearts [33].

### 2.4. Exosomes Secreted by Stem Cells Induce Angiogenesis in the Infarcted Area via NcRNAs

Angiogenesis plays a crucial role in various physiological processes such as wound healing and tissue repair, and promoting vascular formation can significantly aid in the early recovery from cardiac injury. Mesenchymal stem cell-derived exosomes (MSCs-Exo) contain several pro-angiogenic miRNAs, including miR-125a, miR-291, miR-132, miR-17, and miR-210. For example, miR-125a promotes angiogenesis by specifically binding to the 3′-UTR of the angiogenesis inhibitor gene DLL4, thereby inhibiting DLL4 translation [34]. Exosomes secreted under hypoxic conditions were shown to enhance EC angiogenesis and reduce the expression of profibrotic genes in fibroblasts stimulated by TGF-β [35]. However, exosomes derived from different sources of MSCs exhibit variable therapeutic effects. Bone marrow-derived MSCs (BMMSCs), adipose-derived MSCs (ADMSCs), and umbilical cord blood-derived MSCs (UCBMSCs) all promote angiogenesis and inhibit apoptosis by increasing the levels of cytokines such as VEGF, bFGF, and HGF, with ADMSCs-derived exosomes showing the most pronounced effects [36]. These differences are attributed to variations in exosome structure, miRNA content, and the levels of other active proteins. Exosomes isolated from atorvastatin-preconditioned MSCs (MSCsATV-Exo) are rich in LncRNA H19, which upregulates miR-675 in ECs, inducing the secretion of pro-angiogenic factors VEGF and ICAM-1. This promotes cardiac angiogenesis while also suppressing the elevation of IL-6 and TNF-α in the peri-infarct region [37].

## 3. The Role of ncRNA in Treatment Following MI

After MI, the infarcted area experiences prolonged ischemia and hypoxia, leading to the death of numerous normal CMs and causing severe dysfunction of the heart. Furthermore, the array of post-infarction complications and the ischemia–reperfusion injury subsequent to thrombolysis present formidable hurdles. As a crucial cellular signal regulatory factor, NcRNA has the potential to alleviate damage to the infarcted heart. It can stimulate transplanted stem cells to differentiate into normal CMs, thereby replenishing the damaged cardiac tissue, and it exerts cardioprotective effects via exosomes (Table 1 and Figure 2).

### 3.1. The Role of miRNA in MI Repair

An intravenous injection of miR-210 effectively reduces the expression of mitochondrial glycerol-3-phosphate dehydrogenase in CMs, subsequently inhibiting mitochondrial oxygen consumption post-infarction and significantly enhancing glycolytic activity. This process also decreases mitochondrial reactive oxygen species (ROS) levels, with miR-210 overexpression in CMs markedly improving MI and ischemia–reperfusion injury (IRI) outcomes [38]. Additionally, miR-22 reduces p38α levels in CMs, effectively inhibiting apoptosis induced by ischemia–reperfusion in the infarcted heart and promoting cell survival [39]. miR-182/183 contributes to the downstream activation of AKT and ERK pathways by inhibiting RASA1 protein expression, thereby lowering catalase and superoxide dismutase 2 (SOD2) levels in macrophages, improving local inflammatory responses in the infarcted heart and reducing ischemia–reperfusion injury [40].

Notably, miR-218-5p plays a critical role in suppressing post-infarction cardiac fibrosis by inhibiting CX43, which significantly reduces collagen type 1 and α-SMA expression in cardiac fibroblasts [41]. Inhibiting the transcription of miR-143-3p in infarcted CMs using cannabidiol can effectively enhance the expression of Yap and Ctnnd, promote the proliferation of neonatal mouse CMs, and improve cardiac function in infarcted mice [42]. Treating CMs with sevoflurane can slow down the increase in NORAD protein during hypoxia–reoxygenation, increase the content of miR-144-3p in CMs, and protect the myocardium from apoptosis [43]. Knocking down let-7b-5p in neonatal mouse CMs can downregulate the expression of TLR7 and MyD88, thereby reducing the phosphorylation level of NF-κB, inhibiting post-infarction cardiac remodeling, and avoiding heart failure [44]. The overexpression of miR-148a-3p in the myocardium of post-infarction IRI mice can inhibit the expression of the lipid metabolism gene SOCS3, reduce the infiltration of immune cells such as macrophages and monocytes in the infarcted area of the heart after IRI, decrease the levels of inflammatory cytokines IL-1β and TNF-α, and prevent cardiac damage [45]. miR-129-5p inhibits NLRP3 inflammasome activation by downregulating TRPM7, thereby alleviating IRI damage in CMs [46]. Increased USP7 levels in the myocardium of post-infarction rats led to hypoxia-induced cardiomyocyte apoptosis, while the overexpression of miR-409-5p in CMs can effectively downregulate USP7 levels and reduce the expression of IL-1β, TNF-α, and IL-6, improving left ventricular remodeling [47]. Additionally, overexpressing miR-145-5p can inhibit the production of AIFM1 in infarcted CMs, reduce the expression of IL-1β, TNF-α, and IL-6, and promote CM survival [48]. The expression of miR-30e-5p in the left ventricular tissue of rats decreases significantly after MI, and replenishing miR-30e-5p can inhibit PTEN expression, reducing inflammation and myocardial damage caused by MI [49]. The overexpression of miR-708-3p can inhibit the expression of ADAM17 in the myocardium of rats with MI, reducing the production of post-infarction inflammatory cytokines TNF-α, IL-6, and IL-1β, and myocardial injury markers LDH, CK-MB, and cTnI [50]. Similarly, overexpressing miR-26a-5p can also downregulate ADAM17 produced in rat myocardium after hypoxia, reducing the release of creatine kinase-MB and favoring CM survival [51]. Furthermore, miR-26a-5p can reduce the expression level of WNT5A in CMs, inhibit the Wnt/β-catenin signaling pathway, decrease LDH release, and increase SOD and GSH-PX activity, ultimately reducing apoptosis caused by hypoxia reoxygenation after infarction and restoring heart damage [52]. MiR-7a-5p reduced the expression of caspase-3 and Bax in the myocardium by inhibiting the transcription and translation of VDAC1, resisting apoptotic damage caused by IRI [53]. Conversely, knocking down miR-181c-5p in H9C2 cells can enhance the expression of PTPN4 within the cells, reduce LDH and caspase 3 levels, and attenuate hypoxia-reoxygenation-induced myocardial damage [54]. miR-224-5p activated the PI3K/Akt signaling pathway by downregulating PTEN, increasing SOD2 activity, and aiding the survival of CMs under hypoxic conditions after infarction [55]. Rno-miR-30c-5p exhibits significant therapeutic effects on cell inflammation and apoptosis induced by IRI after infarction in rats. This RNA can inhibit SIRT1 and shut down the NF-κB pathway in CMs [56]. Similarly, miR-30e exerts a cardioprotective effect by inhibiting the expression of SOX9 after IRI [57]. An intramyocardial injection of miR-199a-5p provides dual protection to the MI area in male rats. On one hand, miR-199a-5p inhibits the expression of AGTR1, reducing intracellular AngII-induced ROS production and exerting an antioxidant protective effect. On the other hand, miR-199a-5p downregulates MARK4, leading to a decrease in dTyr-tub and enhancing cardiac contractility [58]. miR-322-5p can inhibit the expression of BTG2 protein in the hearts of infarcted rats, shut down the NF-κB signaling pathway, and reduce apoptosis [59]. In H9C2 cells, the overexpression of miR-322-5p downregulates Smurf2, thereby activating the TGF-β/Smad signaling pathway, reducing MI-induced myocardial enzyme production, and alleviating cellular oxidative stress [60,61]. Additionally, miR-322 can induce EZH2 expression and activate the Akt/GSK3β pathway, promoting myocardial repair after infarction [62]. Treatment with dexmedetomidine for IRI damage after infarction increases the content of miR-146a-3p in the heart. This ncRNA downregulates IRAK1 and TRAF6, thereby exerting a cardioprotective effect [63,64]. The overexpression of miR-568 in rat myocardium inhibits the expression of Smurf2, reducing the production of various myocardial enzymes after oxidative stress [65]. Knocking down miR-802-5p in H9C2 cells can promote PTCH1 protein expression, reduce LDH and the production of apoptosis-related Bax protein after hypoxia, and promote cell survival [66]. The overexpression of miR-431 in H9C2 cells can target and inhibit HIPK3, downregulate caspase-3 levels, and reduce the proportion of apoptotic cells, which is beneficial for myocardial survival after infarction [67]. Knocking down miR-327, which is involved in myocardial apoptosis after infarction, can promote the upregulation of ARC protein expression and inhibit the production of pro-apoptotic proteins Fas, FasL, caspase-8, and Bax [68]. The upregulation of miR-21-5p in rat myocardium using PGE1 significantly reduces the content of FASLG, resisting myocardial apoptosis induced by IRI [69].

### 3.2. Role of LncRNA in MI Repair

The heart-specific LncRNA NR_045363, when overexpressed, binds endogenous miR-216a in CMs, attenuating its inhibitory effect on the JAK2-STAT3 pathway. This mechanism stimulates the proliferation of neonatal CMs post-infarction in juvenile mice, significantly improving cardiac function [70]. Conversely, knocking out CRRL in the myocardium of young rats releases miR-199a-3p, suppresses Hopx expression, and promotes post-infarction muscle cell regeneration [71]. LncRNA-Wisper, by binding to TIAR, upregulates Plod2 expression in cardiac fibroblasts; however, upon using ASO to knock down LncRNA-Wisper, fibrosis in the infarcted heart is significantly reduced [72]. Additionally, LncRNA-CAIF directly binds to the p53 protein in CMs, blocking p53-mediated myocardin transcription, ultimately inhibiting cardiac autophagy and mitigating MI damage [73]. Notably, the fibroblast-enriched Cfast LncRNA competitively inhibits COTL1 and TRAP1 binding, enhancing transforming growth factor-beta (TGF-β) signaling and promoting the formation of the SMAD2/SMAD4 complex. Reducing Cfast levels results in markedly improved fibrosis in MI zones [74]. LncRNA UCA1 can act as a competitive endogenous RNA (ceRNA) for miR-128, upregulating the miR-128 target gene SUZ12 and reducing p27 expression, leading to an increase in a series of cell cycle-related protein levels and promoting the proliferation of juvenile CMs. This is crucial for slowing the progression of MI [75]. Additionally, LncRNA UCA1 can also bind to miR-143, releasing the inhibition of miR-143 on MDM2 protein, downregulating p53 levels, and protecting CMs from apoptosis induced by IRI after infarction [76]. LncRNA HOTAIR reduced the expression of apoptosis-related proteins Bax, Bcl-2, and caspase-3 by inhibiting the activity of miR-519d-3p in the myocardium, alleviating cardiomyocyte apoptosis induced by MI or hypoxia [77]. LncRNA growth arrest-specific transcript 5 (GAS5) can exacerbate MI damage by inducing the expression of proteins such as CALM2 [78]. Downregulating LncRNA-GAS5 in the myocardium can promote the inhibitory effect of miR-21 on PDCD4, activate the PI3K/AKT signaling pathway, reduce caspase-9 and BAX levels, and inhibit cardiomyocyte apoptosis after infarction [79]. Similarly, knocking out LncRNA-GAS5 in CMs can release miR-335, thereby inhibiting the expression of ROCK1 and further activating the PI3K/AKT pathway, inhibiting GSK-3β and mPTP. miR-532-5p is also activated after LncRNA-GAS5 knockout, promoting myocardial survival and ultimately improving myocardial apoptotic damage after infarction [80,81]. When catechin is used for MI treatment, the expression of LncRNA MIAT in myocardial tissue decreases, promoting Akt/Gsk-3β activation, thereby improving CM mitochondrial function and alleviating apoptosis in the infarcted area [82]. After infarction, LncRNA LSINCT5 is produced abundantly in CMs, and knocking out LncRNA LSINCT5 can activate the inhibited miR-222, promote PI3K/AKT pathway activation, and favor the survival of the infarcted myocardium [83]. Downregulating LncRNA XIST can promote the inhibitory effect of miR-130a-3p on XIST and PDE4D in the myocardium, reducing CM apoptosis and reversing PDE4D-induced myocardial damage [84]. miR-101a-3p is also activated after silencing LncRNA XIST, inhibiting FOS production in CMs after infarction and resisting apoptosis [85]. Downregulating the level of LncRNA NEAT1 in the myocardium after infarction can promote the inhibition of Atg12 expression by miR-378a-3p, which is a marker of autophagy, reducing myocardial apoptosis in the infarcted area [86]. LncRNA HULC, as an antagonist of miR-377-5p, can significantly inhibit the infarction damage exacerbated by miR-377-5p. After supplementing LncRNA HULC in the infarcted hearts of rats, the expression of NLRP3, Caspase-1, and IL-1β proteins induced by miR-377-5p is downregulated, and the myocardial apoptosis rate is reduced [87]. LncRNA Rian, after binding to miR-17-5p, can reverse the inhibition of miR-17-5p on CCND1 expression in the myocardium, thereby downregulating caspase-1 and GSDMD levels, and avoiding IRI damage after infarction [88].

### 3.3. The Role of ncRNA in Stem Cell-Induced CM Differentiation and Functional Myocardium Supplementation in MI Therapy

The loss of functional myocardium following MI exacerbates cardiac workload, accelerating disease progression. In this context, ncRNAs exhibit significant potential in modulating cellular differentiation and proliferation, promoting stem cell differentiation into CMs and increasing cell survival rates, thereby effectively replacing lost CMs and improving cardiac function. Since 5-Azacytidine was discovered to induce MSC differentiation into CMs-like cells in vitro [89], differential expression of ncRNAs involved in this process has become a research hotspot. These ncRNAs play essential roles in the regulation of MSC differentiation into CMs, with the overexpression or knockdown of these RNAs in cells significantly accelerating MSC differentiation.

Dai et al. observed that miR-199b-5p expression decreased progressively during BMSC differentiation into CM-like cells. The suppression of miR-199b-5p activated the HSF1/HSP70 signaling pathway, significantly upregulating cardiac-specific gene expression [90]. Additionally, miR-124 binds to the 3′ UTR of the STAT3 gene, thereby inhibiting the expression of key cardiac marker proteins (e.g., ANP, TNT, and α-MHC) during MSC–CM differentiation. Introducing antisense oligonucleotide AMO-124 significantly suppressed endogenous miR-124, promoting MSC differentiation into CMs [91]. Another study demonstrated that miR-1a enhances MSC cardiac differentiation induced by 5-azacytidine by reducing the expression of the transcriptional repressor DLL-1 for cardiac genes. During this process, the expression of myocardial marker proteins cTnI, cTnT, and MYH6 increased significantly, and the differentiation duration was considerably shortened [92]. The overexpression of miR-1-2 in mouse BMSCs activated essential molecules in the Wnt/β-catenin signaling pathway, including Wnt11, JNK, β-catenin, and TCF, promoting the upregulation of cardiac-specific genes (e.g., Nkx2.5, cTnI, and GATA4) under 5-azacytidine induction [93]. Overexpressing miR-1 also downregulated the Hes-1 gene, facilitating the expression of early cardiac genes in MSCs cultured in vitro and enhancing the differentiation potential of MSCs into CMs post-transplantation [94].

Furthermore, growing evidence suggests that ncRNAs from other tissues (e.g., adult cardiac tissue) can significantly promote cardiac differentiation when transferred into MSCs. For example, cardiac-inducing RNA (CIR) derived from human adult heart total RNA has been shown to induce various stem cells, including MSCs, ESCs, and iPSCs, to develop into CM-like cells in vitro [95,96]. LncRNA-CIR-6 stimulates MSC cardiac differentiation via a CDK1-related signaling pathway. In vivo, MSCs overexpressing LncRNA-CIR-6 effectively completed myocardial differentiation, replacing lost cells in the infarcted mouse heart, significantly improving cardiac function and enhancing resistance to MI. Moreover, the administration of an adenoviral vector carrying LncRNA-CIR-6 has shown partial repair effects on cardiac injury, further confirming its potential in cardiac regeneration and repair [96]. Another ncRNA, LncRNA-Bvht, originating from early mesoderm, is essential for activating the core cardiovascular gene network, which is crucial in maintaining the developmental fate of nascent CMs. Experimental results indicate that the transfection of an LncRNA-Bvht overexpression plasmid into MSCs significantly enhances myocardial marker protein expression, including key proteins such as Gata4, Gata6, and Isl-1, during cardiac differentiation [97,98].

CMs generated from iPSC differentiation face challenges such as low cell survival rates and potential tumorigenicity after transplantation [99]. Therefore, exploring ncRNAs that can reinitiate the cell cycle in iPSC-CMs or accelerate their maturation into fully functional CMs has become a key research focus. Studies have shown that the overexpression of miR-590-3p in human-induced pluripotent stem cell-derived CMs (hiPSC-CMs) upregulates a range of cell cycle proteins associated with cell proliferation (e.g., CCND1 and CDK4), thereby inducing CM proliferation. The transplantation of hiPSC-CMs overexpressing miR-590-3p into the infarcted heart resulted in a significant increase in surviving cells compared to empty vector controls. Notably, these cells exhibited enhanced anti-apoptotic properties under hypoxic conditions, demonstrating their potential for cardiac repair [100]. Additionally, miR-302b-3p and miR-373-3p promote hiPSC-CM proliferation by inhibiting Lats2 expression, preventing YAP phosphorylation, increasing YAP nuclear translocation, and thus inactivating the Hippo signaling pathway, significantly enhancing cardiac function post-transplantation into the MI area [101]. Similarly, miR-199a overexpression has been shown to increase the proportion of cells in the S and G2/M phases, promoting hiPSC-CM proliferation both in vitro and following transplantation into the infarcted heart, with significant functional recovery observed four weeks post-transplantation [102].

It is noteworthy that while miR-1 overexpression in hiPSCs does not directly induce CM differentiation, more beating CMs are generated when miR-1 overexpression is combined with a cardiac differentiation strategy that includes BMP4, bFGF, and Activin A. On the fourth day of differentiation, miR-1-overexpressing iPSCs displayed increased levels of the mesoderm marker MESP1 and inhibited the expression of the endoderm marker FOXA2 until the 20th day. Moreover, miR-1 overexpression rapidly downregulated the pluripotency factor OCT4 during differentiation. Further studies suggest that miR-1′s pro-cardiomyogenic effect depends on the inhibition of the Wnt and FGF signaling pathways, and the continuous activation of these pathways post-day-four of differentiation, which otherwise impedes CM differentiation, was reversed by miR-1 overexpression [103]. Compared to adult human CMs, hiPSC-CMs retain fetal-like characteristics in size, morphology, ultrastructure, ion channel function, and metabolic features, including being smaller mononuclear cells with fewer mitochondria. However, miRNA induction methods have been shown to effectively mature hiPSC-CMs into adult-like CMs, specifically accelerating maturation by the miR-302 family, let-7 miRNA, miR-133a/b, miR-208a/b, miR-143, and miR-145, which enhances their contractile capability and accelerates the formation of a widespread T-tubule network [104].

### 3.4. Exosomes Secreted by Stem Cells Mitigate Cardiac IRI via ncRNAs

Traditional therapies for MI, such as thrombolysis, although effective at reopening occluded vessels, may induce IRI, further exacerbating cardiac conditions. However, MSC-derived exosomes, as an emerging therapeutic modality, exhibit significant cardioprotective potential. These exosomes can deliver functional ncRNAs to the damaged myocardium, effectively promoting cardiac function recovery by inhibiting apoptosis, alleviating oxidative stress, and preventing fibrosis. When combined with traditional MI therapies, exosome applications offer a novel strategy for the comprehensive mitigation of cardiac injury.

Specifically, several miRNAs within exosomes play key roles in these processes. For instance, miR-125b exerts a potent anti-apoptotic effect by suppressing the expression of pro-apoptotic genes P53 and BAK1 in CMs [105]. Similarly, miR-21 downregulates pro-apoptotic proteins such as caspase-3 and programmed cell death protein 4 (PDCD4), providing strong anti-apoptotic effects. Under oxidative stress conditions, such as in hydrogen peroxide-treated MSCs, miR-21 accumulates in exosomes, suppressing PTEN expression and activating the PI3K/AKT signaling pathway, thus forming a protective mechanism against oxidative stress-induced apoptosis and enhancing CM survival [106]. Additionally, miR-25-3p directly reduces the levels of pro-apoptotic genes FASL and PTEN, while also decreasing EZH2 and H3K27me3 levels, leading to the activation of cardioprotective genes such as eNOS and anti-inflammatory genes like SOCS3, providing further cardioprotective effects [107]. Exosomal miR-19 from MSCs promote CM survival by inhibiting the expression of phosphatase and tensin homolog (PTEN), and activating the AKT/ERK pathway [108]. Meanwhile, miR-210 reduces apoptosis by downregulating apoptosis-inducing factor 3 (AIFM3) and phosphorylated protein kinase B (PKB) in CMs [109]. Notably, the synergistic action of miR-206 and miR-216b inhibits the expression of Atg13, reducing autophagy activation under hypoxia and MI conditions [110]. Moreover, exosomes from hMSCs transfected with LncA2M-AS1 are enriched in LncRNA LncA2M-AS1, which promotes XIAP expression by inhibiting miR-556-5p in CMs, effectively combating apoptosis induced by IRI following MI [111]. The miR-486-5p in BMSC-derived exosomes can inhibit the expression of PTEN in H9C2 cells, activate the PI3K/AKT pathway, and resist apoptosis in H9C2 cells induced by IRI [112]. Additionally, BMSC-derived exosomes can replenish the decreased miR-125b in CMs after IRI. miR-125b directly inhibits SIRT7 protein expression, leading to the downregulation of Bax and caspase-3, the upregulation of Bcl-2, and the reduction in IL-1β, IL-6, and TNF-α levels. Restoring normal levels of miR-125b in the myocardium can effectively prevent myocardial apoptosis and alleviate IRI damage [113]. IRI induces an increased expression of FOXO1 and caspase 3 in rat myocardium. Increasing the content of miR-183-5p in CMs using BMSC-derived exosomes can effectively inhibit FOXO1 protein expression, reduce apoptosis and oxidative stress in CMs induced by IRI, and improve heart function [114]. The miR-144 contained in MSC-derived exosomes can inhibit PTEN expression, increase p-AKT levels, and protect CMs from apoptosis caused by hypoxia, thereby reducing heart damage [115].

In addition to directly inhibiting apoptosis, MSC-derived exosomal miRNAs also provide cardioprotection by mitigating oxidative stress in cardiac cells. Oxidative stress refers to an imbalance between oxidants and antioxidants, favoring oxidation and generating a series of detrimental effects via free radicals. This oxidative damage is particularly severe under hypoxia/reoxygenation conditions. For example, miR-23a-3p inhibits CM ferroptosis by suppressing the expression of divalent metal transporter 1 (DMT1) [116], while miR-214 reduces ROS production and enhances the proliferation of H9c2 cells following hydrogen peroxide stimulation. Additionally, exosomal miR-21 activates the AKT/mTOR pathway in H9c2 cells, inhibiting autophagy and reducing apoptosis induced by hypoxia/reoxygenation, thus suppressing oxidative stress [117]. miR-101 in CMs alleviates hypoxia/reoxygenation injury by inhibiting the expression of DNA damage inducible transcript 4 (DDIT4), thereby preventing autophagy under oxidative stress conditions [118]. miR-143-3p, by downregulating CHK2 gene expression, inhibits hypoxia/reoxygenation-induced autophagy through the CHK2-Beclin2 pathway, effectively reducing cell apoptosis [119]. Notably, MSC-derived exosomes with high CD63 expression show increased levels of miR-29 and miR-24, while miR-34, miR-130, and miR-378 levels are reduced. Injecting CD63-enriched exosomes into the infarcted hearts of SD rats significantly reduces the percentage of fibrotic areas and the total perimeter of fibrosis in cardiac cross-sections [120]. Additionally, macrophage migration inhibitory factor (MIF) promotes MSC-derived exosome-mediated cardiac repair by upregulating miR-133a-3p, reducing the fibrosis area in infarcted hearts [121]. In rats with myocardial IRI, the level of miR-98-5p decreases in myocardial tissue, leading to the inactivation of the PI3K/Akt signaling pathway. The use of BMSC-derived exosomes can replenish the deficient miR-98-5p in the myocardium, reactivate the PI3K/Akt pathway, and subsequently downregulate the level of TLR4. Ultimately, this effectively inhibits the production of myocardial enzymes in the IRI myocardial tissue and reduces oxidative stress damage [122].

Artificially optimizing the composition of MSC-derived exosomes can further enhance their therapeutic efficacy. For instance, delivering exosomes from MSCs overexpressing miR-19a/19b to infarcted myocardium significantly improves cardiac function recovery and reduces cardiac fibrosis [123]. miR-129-5p in MSC-derived exosomes downregulates TRAF3 in CMs, inhibiting the NF-κB signaling pathway and ameliorating ventricular dysfunction in heart-failure mice, while also reducing cardiac fibrosis [124]. Furthermore, exosomes from MSCs transfected with miR-133 agomir show an increased miR-133 content, which suppresses the expression of snail 1 in CMs, effectively preventing left ventricular fibrosis after MI [125]. Genetically engineered hiPSCs and MSCs overexpressing miR-1 and miR-199a can produce exosomes with anti-inflammatory and anti-cardiac fibrosis effects. As the primary functional component, miR-1 can reduce the levels of inflammatory cytokines CCL2 and IL-8 in cardiac fibroblasts and downregulate the expression of the fibrosis-promoting gene α-SMA. Meanwhile, miR-199a-3p reduces myocardial fibrosis by targeting and inhibiting SERPINE2 [126]. Hypoxia-induced iPSCs produce exosomes enriched with miR-302b-3p, which significantly decrease the expression of SMAD2 and TGFBR2 in cardiac fibroblasts, inhibit collagen deposition, and reduce the stiffness of activated fibroblasts, ultimately resisting cardiac fibrosis [127].

**Table 1 ijms-26-00231-t001:** NcRNAs involved in MI treatment.

ncRNA	Function	Pathway	Model	Reference
miRNA				
miR-210	Inhibits mitochondrial oxygen consumption, increases glycolysis, reduces ROS, and improves ischemia–reperfusion injury	Downregulates mitochondrial glycerol-3-phosphate dehydrogenase expression	Mouse myocardial infarction ischemia–reperfusion model	[38]
miR-22	Reduces p38α in cardiomyocytes, inhibits apoptosis, and promotes cell survival	Inhibits p38α	Mouse myocardial infarction ischemia–reperfusion model	[39]
miR-182/183	Promotes Akt and ERK activation, decreases catalase and SOD2 in macrophages, reduces local inflammation and ischemia–reperfusion injury	Inhibits RASA1, activates Akt/ERK	Mouse myocardial infarction model	[40]
miR-218-5p	Inhibits CX43, reduces fibrosis, and significantly lowers collagen type 1 and α-SMA expression	Inhibits CX43/α-SMA	Mouse myocardial infarction model	[41]
miR-143-3p	Stimulates cardiomyocyte proliferation and improves cardiac function	Upregulates Yap/Ctnnd	Juvenile mouse myocardial infarction model	[42]
miR-144-3p	Inhibits apoptosis	Downregulates NORAD	Mouse myocardial infarction model	[43]
let-7b-5p	Aggravates myocardial infarction	Upregulates TLR7/MyD88	Juvenile mouse myocardial infarction model	[44]
miR-148a-3p	Reduces local inflammation	Inhibits SOCS3/IL-1β/TNF-α	Mouse myocardial infarction model	[45]
miR-129-5p	Reduces local inflammation and ischemia–reperfusion injury	Downregulates TRPM7/NLRP3	Mouse myocardial infarction ischemia–reperfusion model	[46]
miR-409-5p	Inhibits apoptosis	Downregulates USP7/IL-1β/TNF-α	Mouse myocardial infarction model	[47]
miR-145-5p	Inhibits apoptosis	Inhibits AIFM1/IL-1β/TNF-α	Mouse myocardial infarction model	[48]
miR-30e-5p	Reduces local inflammation	Downregulates PTEN	Mouse myocardial infarction model	[50,58]
miR-708-3p	Reduces local inflammation	Downregulates ADAM17	Mouse myocardial infarction model	[51]
miR-26a-5p	Inhibits apoptosis	Downregulates ADAM17, inhibits Wnt/β-catenin	Mouse myocardial infarction ischemia–reperfusion model	[52,53]
MiR-7a-5p	Inhibits apoptosis	Downregulates VDAC1, inhibits caspase-3/Bax	Mouse myocardial infarction ischemia–reperfusion model	[54]
Rno-miR-30c-5p	Inhibits apoptosis, reduces local inflammation	Downregulates SIRT1, inhibits NF-κB	Mouse myocardial infarction ischemia–reperfusion model	[57]
miR-199a-5p	Enhances heart contractility	Downregulates MARK4/dTyr-tub	Mouse myocardial infarction model	[59]
miR-322-5p	Inhibits apoptosis	Inhibits BTG2/NF-κB, activates EZH2/Akt/GSK3β	Mouse myocardial infarction model	[60,61,62,63]
miR-146a-3p	Improves cardiac function	Downregulates IRAK1/TRAF6	Mouse myocardial infarction ischemia–reperfusion model	[64,65]
miR-327	Promotes apoptosis	Upregulates ARC/Fas/FasL/caspase-8/Bax	Mouse myocardial infarction model	[68]
miR-21-5p	Inhibits apoptosis	Inhibits FASLG	Mouse myocardial infarction ischemia–reperfusion model	[69]
LncRNA				
NR_045363	Stimulates cardiomyocyte proliferation and improves cardiac function	Inhibits miR-216a’s effect on JAK2-STAT3 pathway	Juvenile mouse myocardial infarction model	[70]
CRRL	Promotes cardiac regeneration and function	Inhibits miR-199a-3p, represses Hopx	Juvenile rat myocardial infarction model	[71]
LncRNA-Wisper	Improves myocardial fibrosis	Upregulates Plod2 via TIAR binding	Mouse myocardial infarction model	[72]
LncRNA-CAIF	Inhibits cardiac autophagy, alleviating myocardial infarction injury	Blocks p53-mediated myocardin transcription	Mouse myocardial infarction model	[73]
Cfast	Improves myocardial fibrosis	Inhibits COTL1/TRAP1, enhances TGF-β/SMAD signaling	Mouse myocardial infarction model	[74]
LncRNA UCA1	Stimulates cardiomyocyte proliferation, inhibits apoptosis	Reduces p27/p53	Mouse myocardial infarction model	[75,76]
LncRNA HOTAIR	Inhibits apoptosis, improves cardiac function	Downregulates Bax/Bcl-2/caspase-3	Mouse myocardial infarction model	[77]
LncRNA-GAS5	Promotes apoptosis	Inhibits PI3K/AKT, upregulates caspase-9/BAX	Mouse myocardial infarction model	[78,79,80,81]
LncRNA MIAT	Promotes apoptosis	Inhibits Akt/Gsk-3β	Mouse myocardial infarction model	[82]
LncRNA LSINCT5	Inhibits cardiomyocyte survival	Inhibits PI3K/AKT	Mouse myocardial infarction model	[83]
LncRNA XIST	Promotes apoptosis	Upregulates XIST/PDE4D, increase FOS	Mouse myocardial infarction model	[84,85]
LncRNA NEAT1	Stimulates cardiac autophagy	Upregulates Atg12	Mouse myocardial infarction model	[86]
LncRNA HULC	Promotes cardiac function, Inhibits apoptosis	Downregulates NLRP3/Caspase-1/IL-1β/GSDMD	Mouse myocardial infarction model	[87,88]

## 4. The Role of ncRNA in End-Stage Treatment of MI

In the end stage of MI, heart damage becomes difficult to reverse, and heart transplantation becomes the only option. Transplanting an allogeneic heart may trigger rejection by the body. The use of immunosuppressive drugs such as everolimus and calcineurin inhibitors can reduce post-transplant rejection and improve patient survival rates [128]. Utilizing ncRNA to induce stem cells to exert immunomodulatory effects can effectively alleviate the inflammatory response after heart transplantation, with lower side effects compared to pharmacological treatment (Table 2).

### Immunomodulatory Effects of ncRNA on Stem Cells in MI Therapy

The interaction between MSCs and damaged cells can alleviate inflammatory responses and significantly alter immune cell phenotypes within the microenvironment of damaged tissue, thereby effectively reducing inflammation. This process, independent of the myocardial differentiation capacity of MSCs, nonetheless promotes cardiac functional recovery. In the advanced stages of MI, heart transplantation often becomes the last viable option for life-saving intervention. However, acute immune rejection following surgery can be effectively suppressed by IL-10, a cytokine released by M2 macrophages [129].

Notably, research has shown that anti-miRNA-204-3p can upregulate CXCR4 (CXC motif chemokine receptor 4) expression in MSCs. This upregulation not only enhances MSC proliferation and migration, but also induces polarization of cardiac macrophages toward an anti-inflammatory M2 phenotype, further reducing local inflammatory responses [130]. The immunomodulatory potential of MSCs is activated by the combined stimulation of inflammatory cytokines IFN-γ, TNF-α, and IL-1β, with these cytokine levels generally correlating positively with inducible nitric oxide synthase (iNOS) activity. Interestingly, the downregulation of miR-155 promotes TAB2 protein expression, which in turn increases iNOS levels within MSCs, thus supporting their full immunoregulatory function [131].

Moreover, the overexpression of miR-19a/19b significantly enhances MSC survival in the MI area by reducing the number of CD68-positive inflammatory cells in the myocardium and lowering the levels of inflammatory cytokines such as IL-1β, TNF-α, IL-6, and MCP-1, leading to marked improvements in post-MI cardiac function recovery in mice [132]. Additionally, miR-335 plays a critical role in regulating the transition of human MSCs (hMSCs) from a resting to a reparative phenotype. IFN-γ-induced downregulation of miR-335 reduces RUNX2 protein expression, facilitating the shift of hMSCs to a reparative phenotype, and activating their intrinsic immunomodulatory properties [133].

These findings not only deepen our understanding of MSC roles in cardiac repair and immune modulation, but also provide novel perspectives and strategies for the treatment of cardiac diseases such as MI.

**Table 2 ijms-26-00231-t002:** NcRNA with immunomodulatory effects.

ncRNA	Function	Pathway	Model	Reference
miRNA				
miRNA-204-3p	Inhibits MSC proliferation, aggravates inflammation	Downregulates CXCR4,inhibits M2 polarization	MSCs and macrophage co-cultures model	[130]
miR-155	Inhibits MSC immunoregulation	Downregulates TAB2, represses iNOS	In vitro inflammation model	[131]
miR-19a/19b	Inhibits inflammation, improves heart function	Downregulates Il-1β, TNF-α, IL-6 and MCP-1	Mouse myocardial infarction model	[132]
miR-335	Inhibits hMSC repair phenotypic switching	Upregulates Runx2	In vitro inflammation model	[133]

## 5. Conclusions

NcRNAs, a class of RNA that does not directly contribute to protein synthesis, play a crucial role in regulating cellular fate. They not only directly influence cardiac vessel regeneration and modulate MI repair, but also participate in the internal regulatory mechanisms of stem cells, guiding their differentiation into CMs and promoting angiogenesis. Moreover, as the key functional components of stem cell-derived exosomes, ncRNAs enhance myocardial cell survival in infarcted areas through paracrine signaling, thereby slowing disease progression (Figure 3).

In summary, developing targeted treatment strategies for different stages of MI underscores the potential of ncRNAs and stem cells to drive revolutionary breakthroughs in future medical care.

## Figures and Tables

**Figure 1 ijms-26-00231-f001:**
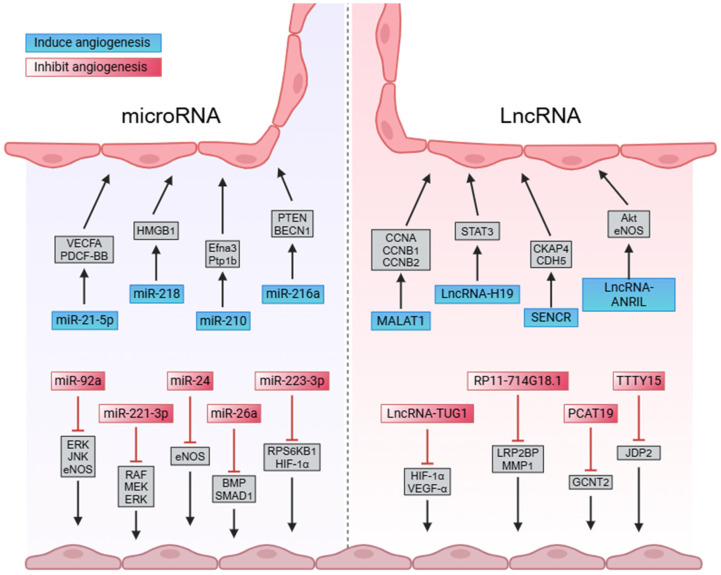
NcRNAs directly affect angiogenesis in the heart.

**Figure 2 ijms-26-00231-f002:**
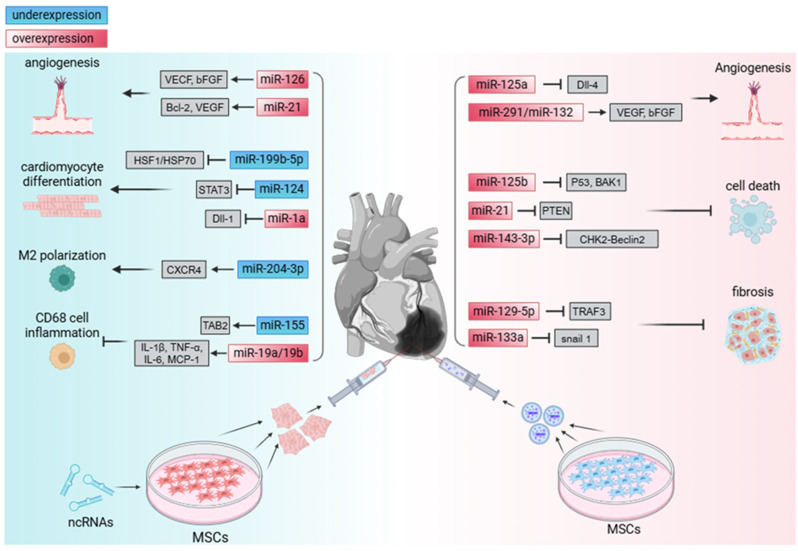
The statistical mechanisms of ncRNAs in stem cell and exosome therapies.

**Figure 3 ijms-26-00231-f003:**
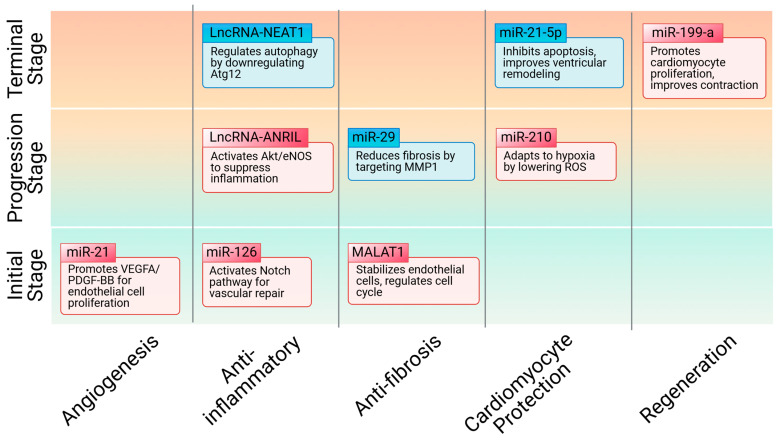
Summary of the ncRNAs involved in different stages of MI treatment.

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
