# Peer review of "Innovative Therapeutic Strategies for Myocardial Infarction Across Various Stages: Non-Coding RNA and Stem Cells"

_ijms, 2024, doi:10.3390/ijms26010231_

Round 1
Reviewer 1 Report
Comments and Suggestions for Authors
The manuscript by Zhuang aims to provide a review on the role of non-coding RNAs as emerging approach to treat myocardial infarction (MI). While the manuscript is potentially interesting, at present it provides a rather unstructured compendium of the role of distinct non coding RNAs in MI. There are two mayor concerns. The first one is that it is unclear which is the target of the review, the non-coding RNAs in MI at large, their role in stem cell therapy in MI or their role in exosomes in MI. It seems that they tried to merge everything together but the link between each part is very poor and thus it should be re-edited. The second point is that a large part of the data presented are just listed on each subheading, with little data integration and lacking a short summary sentence that might ease the reader to digest all the previously mentioned information. Thus, it is compulsory that the authors would make an effort to easy the readership adding some concluding remarks on each of the subheading.
Minor comments
Subtitle in 2.1 and 2.2 are identical. I guess 2.2 refers to lncRNAs. Please modify accordingly.
Subtitle 2 is wrongly written; i.e. Regeneration
Author Response
Comments1.The first one is that it is unclear which is the target of the review, the non-coding RNAs in MI at large, their role in stem cell therapy in MI or their role in exosomes in MI. It seems that they tried to merge everything together but the link between each part is very poor and thus it should be re-edited.
Response1. Thank you for your suggestion and according to your question, we have revised the central focus of the article to emphasize the roles played by ncRNA at different stages of myocardial infarction (MI). From the initial to the final stages of MI, ncRNA either directly acts on the heart or is used in combination therapy with stem cells. This allows us to analyze ncRNA-based treatment approaches tailored to different stages of MI. Please see below.
Comments2.The second point is that a large part of the data presented are just listed on each subheading, with little data integration and lacking a short summary sentence that might ease the reader to digest all the previously mentioned information. Thus, it is compulsory that the authors would make an effort to easy the readership adding some concluding remarks on each of the subheading.
Response2. Thank you for your suggestion and according to your question, At the beginning of each section, we have provided a general overview of the disease characteristics at each stage of MI and the corresponding ncRNA therapeutic approaches, facilitating easier reading and understanding. Please see below.
Reviewer 2 Report
Comments and Suggestions for Authors
The paper by Zhuang B. et al. reviewed the implication of ncRNAs in myocardial infarction recovery, the role of stem cells and the involvement of exosome therapy, trying to provide new therapies against myocardial infarction damage. Although the objective of the review is interesting, the organization and sections of the article make it difficult to read, it is not clear and, in addition to having errors in the titles, it does not follow a correct order, it is not broken down by types of ncRNAs.
There are several recent reviews that include studies related to the present review, although the topic is relevant and interesting, it is not something new. Thus, the following works could be reviewing what is included in this review: PMID: 34718448 and PMID: 32215566.
Furthermore, reviewing the 79 references included, it is estimated that 44% of the articles are not recent (more than 5 years). Furthermore, consulting databases from the last 5 years, more than 1,500 articles referring to ncRNAs and myocardial infarction have been published, so 79 references does not seem to be a very large number for a review of the topic.
The figures presented in the article are appropriate, they also represent the data collected from the references, and are easy to understand and interpret. However, a summary table indicating the different microRNAs and lcRNAs included is missing. For this purpose, authors are recommended to take as a reference the table that appears in the following article PMID: 34502068.
The conclusions are scarce, seem superficial and do not specify which ncRNAs are most relevant in cardiac regeneration after a myocardial infarction and also the regeneration mechanisms in which these ncRNAs are implicated.
Author Response
Comments1. The paper by Zhuang B. et al. reviewed the implication of ncRNAs in myocardial infarction recovery, the role of stem cells and the involvement of exosome therapy, trying to provide new therapies against myocardial infarction damage. Although the objective of the review is interesting, the organization and sections of the article make it difficult to read, it is not clear and, in addition to having errors in the titles, it does not follow a correct order, it is not broken down by types of ncRNAs.
Response1. Thank you for your suggestion and according to your question, We have revisited the title and the content expression of the article, revised inappropriate statements, and reorganized the information regarding the types of ncRNA and their roles in MI treatment. Please see below.
Comments2. There are several recent reviews that include studies related to the present review, although the topic is relevant and interesting, it is not something new. Thus, the following works could be reviewing what is included in this review: PMID: 34718448 and PMID: 32215566.
Response2. Thank you for your suggestion and according to your question, after consulting the articles with PMID: 34718448 and PMID: 32215566, we have adjusted the focus of the article. Now, it emphasizes organizing the therapeutic effects of ncRNA according to different stages of MI: early, middle, and late stages. The theme of the article has become more novel and innovative. Please see below.
Comments3. Furthermore, reviewing the 79 references included, it is estimated that 44% of the articles are not recent (more than 5 years). Furthermore, consulting databases from the last 5 years, more than 1,500 articles referring to ncRNAs and myocardial infarction have been published, so 79 references does not seem to be a very large number for a review of the topic.
Response3. Thank you for your suggestion and according to your question, the 79 references were indeed inadequate. After reading more literature, we have added more comprehensive content to the article, now we have 133 references and the selected references are mainly from the past five years. Please see below.
Comments4. The figures presented in the article are appropriate, they also represent the data collected from the references, and are easy to understand and interpret. However, a summary table indicating the different microRNAs and lcRNAs included is missing. For this purpose, authors are recommended to take as a reference the table that appears in the following article PMID: 34502068.
Response4. Thank you for your suggestion and according to your question, after consulting PMID: 34502068, we supplemented summary tables for each section, compiling the functions and mechanisms of different microRNAs and lncRNAs.Please see below.
Comments5. The conclusions are scarce, seem superficial and do not specify which ncRNAs are most relevant in cardiac regeneration after a myocardial infarction and also the regeneration mechanisms in which these ncRNAs are implicated.
Response5. Thank you for your suggestion and according to your question, we have supplemented and organized the signaling pathways involving NcRNA in MI treatment in the mechanism diagram and table. Among them, the ncRNA that have been extensively studied are highlighted in the main text, as they may have more connections with heart regeneration. These ncRNA are also marked in the conclusion figure for emphasis. Please see below.